# Evaluation of PD-L1 Expression and Anti-*EGFR* Therapy in *EGFR*-Mutant Non-Small-Cell Lung Cancer

**DOI:** 10.3390/medicina61081467

**Published:** 2025-08-15

**Authors:** Gizem Teoman, Elanur Karaman, Şafak Ersöz, Sevdegül Aydın Mungan

**Affiliations:** 1Faculty of Medicine, Department of Medical Pathology, Karadeniz Technical University, Trabzon 61080, Turkey; ersozs@yahoo.com (Ş.E.); drsevdegul@gmail.com (S.A.M.); 2Faculty of Medicine, Department of Medical Oncology, Karadeniz Technical University, Trabzon 61080, Turkey; drelanurkaraman@gmail.com

**Keywords:** NSCLC, *EGFR*, PD-L1, tyrosine kinase inhibitor, prognosis, targeted therapy

## Abstract

*Background and Objectives*: Non-small-cell lung cancer (NSCLC) often has epidermal growth factor receptor (*EGFR*) mutations, which are key targets for therapy. *EGFR* mutation subtypes, especially exon 19 deletions and exon 21 L858R mutations, influence responses to *EGFR* tyrosine kinase inhibitors (TKIs) and patient survival. Despite progress in TKI treatments, resistance and different responses remain challenges. This study explores the relationship between *EGFR* mutation subtypes, PD-L1 expression, and patient outcomes after anti-*EGFR* therapy. *Materials and Methods*: We studied 176 cases of *EGFR* mutation-positive NSCLC. Next-generation sequencing was used to analyze *EGFR* and other mutations, while PD-L1 expression was evaluated through immunohistochemistry. We analyzed *EGFR* mutation subtypes, PD-L1 status, treatments, and survival outcomes. *Results*: Among 176 cases, 88.6% were adenocarcinomas. Within the *EGFR* mutation spectrum, exon 19 deletions were the most common subtype, accounting for 40.9% of cases, followed by the point mutation in exon 21, which occurred in 35.8% of cases. Less frequent alterations, making up 23.3% of all detected mutations, included mutations in exon 18, insertions, and point mutations such as S768I and T790M in exon 20, as well as changes in exon 2, exon 7, and other less frequently affected regions. Exon 19 mutations were associated with older age, female sex, adenocarcinoma, and bone metastasis (*p* < 0.05). *TP53* was the most common concurrent mutation (44.3%). PD-L1 positivity (TPS ≥ 1%) was observed in 48.3%, with high expression (TPS ≥ 50%) in 25.9%. Exon 21 mutations were significantly linked to PD-L1 negativity (*p* = 0.008). The median overall survival was longest with TKI therapy (51 months), and this was also observed in PD-L1-positive patients, although the difference was not statistically significant. The median progression-free survival for patients treated with TKIs and those with *EGFR* mutations was 14 months. PD-L1-positive patients receiving TKIs had significantly longer survival than those who did not (51 vs. 17 months, *p* = 0.003). *Conclusions*: *EGFR* mutation subtypes and PD-L1 expression seem to affect treatment outcomes and survival in NSCLC. The observed links emphasize the potential value of combining molecular and immunological markers to guide therapy choices.

## 1. Introduction

Lung cancer remains one of the most common malignancies worldwide and is the leading cause of cancer-related death globally. Non-small-cell lung cancer (NSCLC) makes up about 85% of lung cancer cases. Despite advances in early detection, only about 25–30% of NSCLC patients are diagnosed at early stages and are eligible for potentially curative treatments such as surgery, with or without additional chemotherapy and/or radiotherapy [1]. However, the overall outlook stays poor due to the high rate of diagnosis at advanced stages.

Epidermal growth factor receptor (*EGFR*), a member of the ErbB receptor tyrosine kinase family, plays a vital role in NSCLC pathogenesis. *EGFR* mutations, mainly deletions in exon 19 and the L858R point mutation in exon 21, are the most common drivers of mutations in NSCLC, especially among non-smokers and East Asian populations [2,3]. These mutations have predictive and prognostic importance because they confer sensitivity to *EGFR* tyrosine kinase inhibitors (*EGFR*-TKIs), which have revolutionized the treatment of advanced NSCLC [4,5]. Phase III clinical trials have shown better outcomes with *EGFR*-TKIs compared to standard chemotherapy in patients with *EGFR*-mutant advanced NSCLC [6].

Recent evidence has expanded the use of *EGFR*-TKIs into the perioperative setting for early-stage NSCLC. The ADAURA trial demonstrated that adjuvant osimertinib significantly improves disease-free survival in patients with resected *EGFR*-mutant NSCLC, establishing a new standard of care in this field [7]. Furthermore, immunotherapy, especially immune checkpoint inhibitors targeting programmed cell death 1 (PD-1) and its ligand PD-L1, has been integrated into neoadjuvant and adjuvant treatment plans, thereby enhancing long-term outcomes and transforming the management of early-stage NSCLC [8,9].

Immune checkpoint regulation involves co-stimulatory and co-inhibitory molecules that regulate tumor-infiltrating lymphocytes (TILs) [10]. Blocking inhibitory checkpoints with antibodies boosts anti-tumor immunity, mainly by activating CD8+ T cells. Despite the success of PD-1/PD-L1 inhibitors in NSCLC, the interaction between *EGFR* mutations and PD-L1 expression remains complex and not fully understood. Preclinical studies demonstrate that mutant *EGFR* increases PD-L1 expression, whereas *EGFR*-TKI therapy may decrease PD-L1 levels, suggesting a dynamic relationship that affects treatment response [11]. Clinical studies on PD-L1 as a predictive and prognostic biomarker in *EGFR*-mutant NSCLC treated with *EGFR*-TKIs have shown mixed results [12], emphasizing the need for further research.

In this study, we aim to characterize PD-L1 expression in *EGFR*-mutant NSCLC patients and examine the association between *EGFR* mutation subtypes and clinical parameters, including age, sex, histologic subtype, first-line treatment, overall survival, and progression-free survival.

## 2. Materials and Methods

### 2.1. Study Population

The DNA profiles of 176 cases diagnosed with non-small-cell lung carcinoma between January 2020 and October 2024 were analyzed by the Department of Pathology at the Faculty of Medicine, Karadeniz Technical University, using next-generation sequencing (NGS). The cases included in this study were diagnosed with non-small-cell lung carcinoma (adenocarcinoma or squamous cell carcinoma) in our department between January 2020 and January 2024 and were found to have EGFR mutations based on DNA gene profiling using NGS.

A pathologist specialized in thoracic pathology examined hematoxylin- and eosin-stained slides of formalin-fixed paraffin-embedded (FFPE) tissue samples from each patient diagnosed with non-small-cell lung carcinoma and selected the most representative tumor block for further analysis.

Out of 176 cases, diagnosis was confirmed through surgical resection specimens in 148 patients, while the remaining 28 cases were diagnosed using transbronchial EBUS biopsy or CT-guided transthoracic biopsy. Among the 176 patients in the study, 148 underwent surgical resection (such as wedge resection, lobectomy, or pneumonectomy), primarily for diagnostic purposes or due to early-stage disease. The other 28 patients were diagnosed via EBUS or transthoracic biopsy, typically because of advanced-stage disease or inoperability.

PD-L1 immunohistochemistry and molecular analyses were primarily performed on surgical specimens and primary tumor tissue in most cases, as these samples offer a more complete tissue architecture and higher tumor cellularity. When only biopsy material was available, careful selection of samples with sufficient tumor content was made to ensure the accuracy of both PD-L1 immunohistochemistry and next-generation sequencing (NGS) analyses. Therefore, we do not believe that using biopsy specimens in some patients significantly affected the accuracy of PD-L1 scoring or the detection of genetic mutations. This methodological approach aligns with current international guidelines and is supported by published evidence validating the use of biopsy-based assessments in lung cancer diagnostics.

In patients without resectable disease, PD-L1 expression was evaluated using the most accessible metastatic tissue, such as lymph nodes or lung lesions. Importantly, no PD-L1 tests were performed on liver or brain metastases in this study.

Tumor biomarker expression, including EGFR mutations and PD-L1, was independently assessed by two pathologists. Inter-observer variability was evaluated through consensus review, and any disagreements were resolved by joint re-examination to ensure consistent scoring. All immunohistochemical and molecular analyses were performed using standardized protocols and were blinded to clinical data to minimize bias. Cases of non-small-cell lung carcinoma were confirmed through histological examination, and sufficient clinical and pathological data were available for the study. Clinical information was obtained via the hospital system and included patient age, sex, histological type (adenocarcinoma or squamous cell carcinoma), EGFR mutation subtype, PD-L1 expression status, metastatic sites, stage, first-line treatment, tyrosine kinase inhibitor (TKI) therapy, and survival data. This study excluded cases where DNA quality was deemed too low based on DNA profile evaluation.

This study did not receive any specific grants from funding agencies in the public, commercial, or not-for-profit sectors.

### 2.2. DNA Extraction and EGFR Mutation Assay

Molecularly, *EGFR* somatic mutations were analyzed in these cases using the NGS method. The NGS-DNA panels used at different times included the QIAsec solid custom MSI panel, QIAact AIT DNA UMI panel, and AIT Basic QIAact Actionable Insights Tumor Panel. For DNA mutation analysis, genes such as *ALK*, *BCOR*, *BRAF*, *BRCA1*, *BRCA2*, *CDKN2A*, *CTNNB1*, *EGFR*, *ERBB2*, *ERBB3*, *FBXW7*, *FGFR1*, *HRAS*, *IDH1*, *IDH2*, *KIT*, *KRAS*, *MAP2K1*, *MET*, *MLH1*, *MSH2*, *MSH6*, *NRAS*, *NTRK1*, *NTRK2*, *NTRK3*, *PDGFRA*, *PIK3CA*, *PIK3R1*, *PMS2*, *POLE*, *PTEN*, *ROS1*, *RPL22*, *SMAD4*, *TERT*, *TP53*, and *WNT1* were examined. Regarding MSI detection, the loci examined included *BAT40(T)37*, *MONO-27(T)27*, *BAT26(A)27*, *NR24(T)23*, *BAT25(T)25*, *NR22(T)21*, *HSP110-T17(T)17*, *NR21(A)21*, and *BAT34C4(A)18*.

DNA was extracted from paraffin-embedded tumor tissues using the QIAGEN GeneRead DNA/RNA FFPE Kit, Qiagen, Hilden, Germany. The DNA quantity obtained (measured with Qubit 4) was sufficient for the study. DNA quality was assessed by using gel electrophoresis (Qiaxcel method). Next-generation sequencing (NGS) was performed on the Illumina NovaSeq system. Bioinformatics analysis was conducted using the Qiagen Clinical Insight (QCI) Interpret and CLC Genomic Workbench platforms (v. 24.0.3).

In DNA-NGS studies using FFPE materials, variants are identified with a read depth of 500X, an allele frequency of 5% or higher, and a quality score (QUAL) above 200. The detected variants are classified according to Tier Classification, and those with clinical significance are considered pathogenic or likely pathogenic variants.

### 2.3. PD-L1 Expression Analysis

Formalin-fixed paraffin-embedded (FFPE) tissue sections were used to assess PD-L1 expression through immunohistochemistry (IHC) with the SP263 clone (Ventana Medical Systems). PD-L1 expression was measured by the percentage of viable tumor cells showing partial or complete membrane staining. FFPE tumor specimens containing fewer than 100 tumor cells were considered inadequate for PD-L1 analysis. In addition to the Tumor Proportion Score (TPS), PD-L1 expression was also evaluated as a dichotomous variable, categorized as negative (TPS < 1%) or positive (TPS ≥ 1%). This cutoff aligns with international clinical guidelines, including those from the International Association for the Study of Lung Cancer (IASLC), which considers TPS ≥ 1% as a clinically meaningful threshold for PD-L1 positivity in lung cancer [13].

All slides were independently assessed by two experienced pathologists to ensure consistency. No digital pathology tools were utilized in the evaluation. Inter-observer variability was examined through a consensus review, and any disagreements between the pathologists were resolved by joint re-evaluation to enhance scoring reliability.

### 2.4. Systemic Treatment Approach

Initially, systemic chemotherapy was administered to symptomatic patients before the results of next-generation sequencing (NGS) were available for detecting oncogenic driver gene mutations. During this period, platinum-based chemotherapy regimens were the standard first-line treatment. Once the NGS results were obtained, patients with *EGFR* mutations were switched to targeted therapy with tyrosine kinase inhibitors (TKIs). This approach reflects current clinical practice, where delays in molecular diagnostics often result in the initiation of empirical chemotherapy in symptomatic patients. Adjusting therapy based on molecular findings helps optimize treatment effectiveness and patient outcomes. Therefore, the absence of TKI therapy in some *EGFR*-mutant cases should not be viewed as non-compliance with guidelines but as a reflection of real-world treatment dynamics shaped by diagnostic timelines, clinical urgency, and drug availability.

Patients were classified as symptomatic based on their clinical presentation at diagnosis, which included respiratory symptoms (e.g., cough, dyspnea, hemoptysis), weight loss, fatigue, or radiologically significant tumor burden. Additionally, performance status (ECOG) and clinical urgency were considered.

The decision to switch to targeted therapy instead of continuing systemic chemotherapy was made after *EGFR* mutations were identified through NGS. In addition to the presence of actionable mutations, treatment options were based on several clinical factors, including the patient’s response to initial chemotherapy (assessed through imaging and clinical improvement), chemotherapy tolerability, performance status, comorbidities, and physician judgment. Patients who exhibited poor or no response to chemotherapy, or who experienced significant treatment-related toxicity, were prioritized for switching to *EGFR* tyrosine kinase inhibitor (TKI) therapy once *EGFR* mutations were confirmed.

Chemotherapy regimens were selected based on the tumor’s histological subtype and individual patient factors, such as age, performance status, and comorbidities. Doublet chemotherapy protocols included combinations like pemetrexed with cisplatin or carboplatin, paclitaxel with cisplatin or carboplatin, gemcitabine with cisplatin or carboplatin, and docetaxel with cisplatin. For some patients, single-agent chemotherapy regimens such as vinorelbine or docetaxel were used.

All patients with confirmed *EGFR* mutations, including exon 19 deletions and exon 21 L858R mutations, were evaluated for *EGFR* TKI therapy once molecular results were available. The decision to start or switch to *EGFR*-targeted therapy was primarily based on the presence of the actionable mutation but also took into account clinical factors such as response to previous chemotherapy, performance status, comorbidities, and tolerance to systemic therapy. Therefore, patient selection was tailored individually, and not all patients immediately transitioned to TKI therapy after *EGFR* mutation detection. The treatments included erlotinib, gefitinib, afatinib, and osimertinib (the latter in cases with the T790M resistance mutation).

No patient in our cohort received immunotherapy as a first-line treatment. At the time of initial diagnosis and treatment initiation, immunotherapy was not commonly used in patients with actionable *EGFR* mutations, as per national guidelines and standard practices. PD-L1 expression (measured by Tumor Proportion Score, TPS) was assessed retrospectively and was included in survival analysis. When used, immunotherapy was typically introduced in later lines of therapy after TKI resistance or in cases without targetable mutations.

In patients who experienced disease progression under TKI therapy and had an estimated life expectancy of more than three months, additional molecular testing was recommended to identify secondary resistance mutations, including the T790 M mutation. When the T790M mutation was detected, osimertinib was administered as the next line of treatment. It is important to note that although osimertinib has shown clinical benefit as a first-line agent in patients with *EGFR* exon 19 deletions or exon 21 L858R mutations, most patients in this study initially received first- or second-generation TKIs (such as erlotinib, gefitinib, or afatinib). This was due to the retrospective nature of the study and the treatment period, during which osimertinib was not routinely approved or reimbursed as a first-line therapy in the national treatment guidelines. As a result, the use of osimertinib was mainly limited to subsequent lines after detecting resistance mutations. These treatment decisions were made based on drug availability, institutional protocols, and the active national regulatory approvals during the study period.

### 2.5. Statistical Analysis

All data were entered into the SPSS database (IBM SPSS Statistics, version 27.0), and a chi-square test was performed to compare *EGFR* mutation status, the presence of concomitant mutations, and age data obtained through NGS. Descriptive statistics for the results were presented as frequencies and percentages for categorical variables, and as means, standard deviations, minimum values, and maximum values for continuous variables. The normality of continuous variables was assessed using the Kolmogorov–Smirnov test. For comparing continuous variables between two independent groups, Student’s *t*-test was used when data were normally distributed, and the Mann–Whitney U test was used when normality was not satisfied. Overall survival (OS) was calculated from the date of diagnosis to the date of death or to the date of the last follow-up if the patient was still alive. For survival analysis, univariate analysis was performed using the Kaplan–Meier method, and statistical significance was assessed with the log-rank test. Independent prognostic factors for overall survival were evaluated through multivariate Cox proportional hazards regression analysis. The hazard ratio (HR) was used to measure the effect of variables on overall survival. For all variables, 95% confidence intervals (CIs) were calculated and reported when applicable. A *p*-value of less than 0.05 was considered statistically significant.

## 3. Results

This study included a total of 176 cases diagnosed with non-small-cell lung carcinoma (NSCLC). Of these, 156/176 cases (88.6%) were classified as adenocarcinomas, while the remaining 20/176 cases (11.4%) were identified as squamous cell carcinomas (SCCs). The average age of the patients was 66.23 years, with a standard deviation of 10.216, and the age range spanned from 40 to 90 years. A total of 69/176 patients (39.2%) were under 65 years old, while 107/176 (60.8%) were 65 years or older. Regarding sex distribution, 91/176 patients (51.7%) were male, and 85/176 (48.3%) were female.

Patients were divided into early-stage (Stage I–II) and advanced-stage (Stage III–IV) groups. Out of 176 cases, 20/176 (11.4%) were classified as early-stage, while 156/176 (88.6%) were advanced-stage. Lymph node involvement was observed in 22/176 cases (12.5%). All cases were de novo. However, due to incomplete clinical data, smoking history was unavailable for some patients, so a definitive percentage regarding smoking status could not be reported.

Within the *EGFR* mutation spectrum, exon 19 deletions were the most common subtype, found in 72/176 (40.9%) cases. This was followed by the point mutation in exon 21, detected in 63/176 (35.8%) cases. Less common alterations, accounting for 41/176 (23.3%) mutations, included mutations in exon 18, insertions, and point mutations such as S768I and T790M in exon 20, as well as changes in exon 2, exon 7, and other less frequently affected regions. Figure 1 shows the distribution of *EGFR* mutations among the cases.

Among cases with *EGFR* exon 19 mutations, 37/72 (51.4%) were aged 65 or older. A statistically significant association was found between older age and the presence of *EGFR* exon 19 mutations (*p* = 0.033).

Among cases with *EGFR* exon 19 mutations, 46/72 (63.8%) were female, and a statistically significant association was found between female sex and the presence of *EGFR* exon 19 mutations (*p* = 0.001). *EGFR* exon 18 mutations, along with mutations in other exons, were significantly more common in male patients (*p* = 0.014 and *p* < 0.001, respectively).

Among the cases, 156/176 (88.6%) were adenocarcinomas, and 20/176 (11.4%) were squamous cell carcinomas. *EGFR* exon 19 mutations were significantly more common in adenocarcinoma cases (*p* = 0.003).

The distribution of age, sex, histological subtype, concurrent mutations, sites of metastasis, programmed death-ligand1 (PD-L1) expression status, and tumor-infiltrating lymphocyte (TIL) status across *epidermal growth factor receptor* (*EGFR*) mutation subtypes is summarized in Table 1.

In our study, most cases involved bone metastasis in 60/176 patients (34.1%), followed by lymph node metastasis in 22/176 (12.5%), and brain metastasis in 22/176 (12.5%). Notably, bone metastasis was significantly more common in cases with *EGFR* exon 19 mutations (*p* = 0.016). The overall survival for patients with liver metastasis was 14 months, which was shorter compared to patients with bone, brain, or lymph node metastasis. Among metastatic cases, those with lymph node involvement had the longest overall survival, with a median of 51 months.

In this study, 85/176 cases (48.3%) were PD-L1-positive (PD-L1 tumor proportion score ≥ 1%). Among these, 22/85 cases (25.9%) had a PD-L1 TPS (Tumor Proportion Score) of 50% or higher. Exon 19 mutations appeared in 37/85 PD-L1-positive cases (43.5%), exon 20 mutations in 9 cases (10.6%), exon 21 mutations in 22 cases (25.9%), and exon 18 mutations in 8 cases (9.4%). The remaining 9 cases fell into the “other exons” group: 1 involved exon 17, 2 involved exon 7, 1 involved exon 6, 1 involved exon 8, 2 involved exon 15, and 2 involved exon 2. A statistically significant association was found between *EGFR* exon 21 mutations and PD-L1 negativity (*p* = 0.008). A statistically significant association was observed between *EGFR* exon 21 mutations and PD-L1 negativity (*p* = 0.008). No significant correlation was observed between PD-L1 expression and other *EGFR* exon mutation subtypes. Additionally, no significant link was identified between *EGFR* mutation status and the presence of tumor-infiltrating lymphocytes (TILs) or PD-L1 positivity within TILs.

In our study, the median overall survival was 51 months for patients with positive PD-L1 expression, compared to 17 months for those with negative PD-L1 expression (Figure 2). However, this difference did not reach statistical significance (*p* = 0.101). Similarly, no statistically significant relationship was found between the presence of tumor-infiltrating lymphocytes (TILs) and overall survival (*p* = 0.818).

Among the 85 patients with PD-L1 positivity, 52 (61.1%) received tyrosine kinase inhibitor (TKI) therapy. Notably, among PD-L1-positive cases, the median overall survival was 51 months for patients who received TKI therapy, compared to 17 months for those who did not (Figure 3). This difference was statistically significant (*p* = 0.003).

None of the patients in our cohort received immunotherapy as first-line treatment. When it was used, immunotherapy was typically administered later in treatment, either after developing resistance to tyrosine kinase inhibitors (TKIs) or in cases without targetable mutations. A total of 11 patients in our study received immunotherapy. The overall survival of these patients was analyzed statistically. Although the median overall survival was longer for patients who received immunotherapy (30 months) compared to those who did not (21 months), this difference was not statistically significant (*p* = 0.833).

In our study, 58/176 (32.9%) patients received tyrosine kinase inhibitor (TKI) therapy as first-line treatment, with median overall survival of 51 months. Among these 58 patients, 44 received first-generation TKIs (erlotinib or gefitinib), 11 received second-generation TKIs (afatinib), and only 3 patients received third-generation TKIs (osimertinib) as initial treatment. However, additional patients were later switched to osimertinib after the detection of acquired T790M resistance mutations. In total, 19/176 (10.79%) patients ultimately received osimertinib during their treatment course. Statistical analysis showed that patients who received osimertinib had significantly longer overall survival compared to those who did not. The mean overall survival was 92 months in the osimertinib group versus 21 months in the non-osimertinib group (*p* = 0.002).

Chemotherapy was given to 90 patients (51.1% of the total), with median overall survival of 21 months. Of these patients receiving first-line chemotherapy, 4/90 (4.4%) were treated with a single agent, 61/90 (67.8%) received doublet chemotherapy, and the remaining 25/90 (27.8%) were treated with a combination of three or more agents.

In contrast, patients who did not receive any form of systemic treatment (28/176, 16%) had median overall survival of 16 months (Figure 4). Notably, patients with *EGFR* mutations treated with TKIs showed the best survival outcomes, with median overall survival of 51 months (95% CI) (*p* = 0.033). These findings further emphasize the prognostic importance of *EGFR* mutations and highlight the positive effect of TKI therapy on overall survival. The median progression-free survival for cases with *EGFR* mutations was 14 months.

In our study, the Eastern Cooperative Oncology Group (ECOG) performance status at diagnosis was recorded for all patients and included in the overall survival (OS) analysis. ECOG status was not assessed in isolation; instead, it was used as a covariate in a multivariate Cox proportional hazards regression model, along with age, sex, disease stage, and TKI treatment status. This approach enabled us to assess the independent prognostic impact of ECOG status on survival, adjusting for relevant clinical factors.

In the multivariate Cox regression analysis, performance status (ECOG ≥ 2), lack of TKI therapy, and advanced stage at diagnosis were identified as independent predictors of poorer overall survival. Specifically, patients with an ECOG performance status of ≥2 had a 1.72-fold increased risk of death compared to those with an ECOG performance status of 0–1 (HR: 1.718, 95% CI: 1.062–2.779, *p* = 0.027). The absence of TKI treatment was associated with a significantly higher risk of mortality (HR: 3.313, 95% CI: 2.050–5.356, *p* < 0.01). Additionally, patients with advanced-stage disease faced a 2.3-fold increased risk of death compared to those with early-stage disease (HR: 2.319, 95% CI: 1.097–4.903, *p* = 0.028). Gender and age were not significantly associated with survival in the multivariate model. These findings are summarized in Table 2.

In our study, the most common concurrent mutation with *EGFR* was *TP53*, found in 78/176 cases (44.3%). *TP53* was followed by *PIK3CA* in 13/176 (7.4%), *KRAS* in 7/176 (4%), and *PTEN* in 4/176 (2.3%) mutations. *MET* alterations, including amplifications or mutations, were not observed in any cases in our series. The prognostic and predictive significance of *TP53* co-mutations in *EGFR*-mutant NSCLC is well supported by the literature. Among the 78 patients with both *EGFR* and TP53 mutations, 51 (65.3%) received TKI therapy. The average overall survival was 42 months for patients with *TP53* mutations and 49 months for those without, although this difference was not statistically significant (*p* = 0.924).

## 4. Discussion

This study analyzed the clinicopathological features and molecular changes in a cohort of 176 patients diagnosed with NSCLC, focusing on *EGFR* mutations, PD-L1 expression, and their associations with patient outcomes. The results revealed several key insights that improve our understanding of the prognostic factors in NSCLC and the effects of targeted therapies.

The study population primarily consisted of adenocarcinomas (88.6%), with a smaller proportion of squamous cell carcinomas (11.4%). These proportions align with current epidemiological data, which indicate that adenocarcinoma is the most common histological subtype of NSCLC, especially among non-smokers and older adults.

In our cohort, the average age was 66.23 years, with patients ranging from 40 to 90 years old, and most (60.8%) were aged 65 or older. While previous studies have generally reported a higher prevalence of classical *EGFR* mutations among younger patients, our findings reveal a statistically significant association between older age and *EGFR* exon 19 mutations [14]. This difference may result from population-specific factors, including environmental exposures, genetic backgrounds, or biases inherent in retrospective studies. It is also possible that variations in study design, sample size, and regional demographics contribute to these differing results. Therefore, although younger age is often considered characteristic of *EGFR*-mutant NSCLC, our data emphasize the importance of considering broader patient demographics and suggest that *EGFR* mutations, especially exon 19 deletions, may also be common among older populations.

Our findings align with those of Tang et al., who reported a higher prevalence of *EGFR* mutations in patients diagnosed after the age of 50. This supports the idea that older age may significantly influence *EGFR* mutation status in NSCLC patients. It highlights the need to include a broad age range in future studies and to consider age-related biological and environmental factors that could impact mutation rates. Larger, prospective studies are necessary to better understand the complex relationship between age and *EGFR* mutation status across different populations [15].

The distribution of *EGFR* mutations in this study showed that exon 19 deletions were the most common subtype, present in 40.9% of cases, followed by the L858R point mutation in exon 21 at 35.8%. These results align with previous large-scale genomic studies, indicating that exon 19 deletions and exon 21 L858R substitutions together account for approximately 85–90% of activating *EGFR* mutations in NSCLC [16]. Besides these common alterations, less frequent mutations—including those in exon 18, insertions and substitutions in exon 20, and rare mutations in other regions—accounted for 23.3% of all *EGFR*-mutated cases in our study. This highlights the molecular diversity of *EGFR*-mutant NSCLC and emphasizes the increasing recognition of the clinical importance of uncommon *EGFR* mutations, many of which show variable sensitivity to tyrosine kinase inhibitors (TKIs) [17].

A statistically significant association was found between older age and the presence of *EGFR* exon 19 mutations, suggesting that older patients may be more susceptible to this specific alteration. This aligns with research indicating that age-related genomic instability may lead to the accumulation of specific driver mutations in NSCLC [15]. Additionally, a notable connection between female sex and *EGFR* exon 19 mutations was observed, supporting previous findings that *EGFR* mutations, especially exon 19 deletions and L858R substitutions, are more common in female patients, particularly non-smokers [18].

Bone metastases are a common complication in advanced NSCLC, often leading to skeletal-related events and poor clinical outcomes [19]. In our study, liver metastasis was linked to the worst prognosis, with median overall survival of 14 months, aligning with previous evidence showing that liver involvement is an adverse prognostic factor in *EGFR*-mutant NSCLC [20]. Conversely, patients with lymph node metastasis showed the most favorable survival outcomes among metastatic cases, with median overall survival of 51 months. This finding aligns with the existing literature, suggesting that prognoses vary significantly depending on the metastatic site. Specifically, large-scale population-based studies have reported that organ metastases, such as liver involvement, are associated with the shortest overall survival, while lymph node metastasis is linked to better prognostic outcomes [21].

PD-L1 expression is a crucial biomarker in NSCLC, guiding the use of immune checkpoint inhibitors, especially anti-PD-1/PD-L1 therapies [22]. In this study, PD-L1 expression was evaluated using the SP263 clone, a widely employed marker in clinical practice. However, it is essential to acknowledge that variability exists among different PD-L1 immunohistochemistry assays, including SP263, 22C3, 28-8, and SP142 clones. Several studies have shown that, although SP263 and 22C3 clones typically demonstrate good agreement, differences in staining intensity, scoring thresholds, and tumor proportion scoring can impact comparability and clinical interpretation across assays [23]. This variability may limit the direct external applicability of our findings to settings that use alternative PD-L1 assays. Therefore, caution should be exercised when applying PD-L1 expression results obtained with SP263 to broader populations or to studies using different clones. Future research should focus on standardizing PD-L1 testing methods and incorporating cross-validation of assays to enhance comparability and strengthen biomarker-driven therapeutic decision-making.

We found a statistically significant link between *EGFR* exon 21 mutations and PD-L1 negativity, which supports previous research indicating that EGFR-mutant tumors generally exhibit lower PD-L1 expression and fewer immune cells infiltrating them [24]. In contrast, no significant links were observed between PD-L1 expression and other *EGFR* mutation types.

Importantly, among the 85 patients with PD-L1 positivity, 52 (61.1%) received TKI therapy. Subgroup analysis showed that PD-L1-positive patients who received TKI therapy had significantly longer median overall survival (51 months) compared to those who did not receive TKI treatment. These results suggest a potential survival benefit from TKI therapy, even among patients with PD-L1-positive tumors, and raise questions about the interaction between oncogenic driver mutations, targeted therapies, and the immune landscape. This finding may reflect a complex biological context where *EGFR*-targeted therapy has indirect effects on the tumor microenvironment or delays disease progression in molecularly selected subsets of patients.

In our study, none of the patients received immunotherapy as a first-line treatment. Immunotherapy was primarily used in later lines of therapy, often after resistance to *EGFR*-TKIs developed or when no targetable oncogenic drivers were present. A total of 11 patients received immunotherapy. Although these patients showed a longer median overall survival (30 months) compared to those who did not receive immunotherapy (21 months), the difference was not statistically significant. This may be due to the small sample size and the use of immunotherapy in more advanced disease stages, where tumor burden and performance status could have affected survival outcomes. Additionally, the limited effectiveness of immune checkpoint inhibitors in *EGFR*-mutant NSCLC, particularly when used after TKI resistance, has been well documented in previous studies [24]. Despite the trend toward improved survival, our findings suggest that immunotherapy may provide limited benefits in unselected *EGFR*-mutant populations, particularly when given outside of biomarker-driven treatment strategies. Future prospective studies with larger groups and stratified treatment plans are needed to clarify the role of immunotherapy in this subgroup and to find potential biomarkers of response beyond PD-L1 expression.

In our study, patients who received first-line tyrosine kinase inhibitor (TKI) therapy showed a favorable survival outcome, with median overall survival of 51 months. Notably, those who ultimately received osimertinib—either as first-line therapy or after progression due to acquired T790M mutations—demonstrated significantly longer mean overall survival compared to those who did not receive osimertinib. This considerable difference highlights the critical role of osimertinib in extending survival in patients with *EGFR*-mutated NSCLC. Osimertinib, a third-generation *EGFR*-TKI, is highly selective for sensitizing *EGFR* mutations and the T790M resistance mutation and has shown superiority in both progression-free and overall survival compared to earlier-generation TKIs [25]. The pivotal FLAURA trial reported median overall survival of 38.6 months in patients receiving first-line osimertinib, compared to 31.8 months in those receiving first-generation TKIs [4]. Our study further supports these findings, especially when osimertinib is incorporated at any stage during treatment, whether upfront or after resistance develops.

The extended survival seen in patients treated with osimertinib may result from several factors, including proper sequencing of therapies, timely identification of the T790M resistance mutation, suitable patient selection, and tumor biology. Our results highlight the significance of molecular monitoring during disease progression and support the incorporation of osimertinib into treatment plans, either as a first-line therapy or as part of a sequential approach. Real-world data from the literature also show that patients who access osimertinib after T790M positivity experience a notable survival advantage [26].

In our study, ECOG performance status, TKI treatment status, and stage at diagnosis were identified as independent prognostic factors for overall survival in patients with EGFR-mutant NSCLC. Specifically, patients with an ECOG performance status ≥2 had a significantly higher risk of mortality compared to those with an ECOG of 0–1. These findings align with the existing literature emphasizing the strong prognostic value of baseline functional status in advanced lung cancer. Poor performance status has consistently been linked to reduced treatment tolerance, limited access to systemic therapies, and poorer clinical outcomes [27].

Furthermore, the lack of TKI therapy proved to be the most significant independent predictor of poor survival in our cohort. This aligns with strong evidence from numerous randomized clinical trials demonstrating the survival advantage of *EGFR*-targeted therapies in molecularly selected patients with NSCLC [28]. The notable effect of TKI exposure in our real-world population further highlights the importance of prompt molecular testing and proper initiation of targeted therapy.

Additionally, patients diagnosed with advanced-stage disease had a significantly higher risk of death, underscoring the prognostic importance of disease burden at initial presentation. In contrast, neither age nor sex was significantly linked to survival in our multivariate model, indicating that biological and treatment-related factors may be more influential than demographic variables in this context. Overall, our findings highlight the multifactorial nature of prognosis in *EGFR*-mutant NSCLC, supporting a comprehensive approach that incorporates molecular, clinical, and functional parameters to predict patient outcomes and inform treatment plans.

The prognostic and predictive effects of concurrent TP53 mutations in *EGFR*-mutant NSCLC have received increasing focus in recent years. Although our study did not find a statistically significant difference in overall survival between patients with and without TP53 co-mutations, these results should be viewed within the wider context of the existing literature.

Numerous studies have shown that *TP53* co-mutations are linked to poorer clinical outcomes in *EGFR*-mutant NSCLC. Specifically, *TP53* mutations have been associated with shorter progression-free survival (PFS) and overall survival (OS) in patients treated with *EGFR* tyrosine kinase inhibitors (TKIs). For example, Canale et al. reported that *TP53* mutations are independent predictors of poor prognosis in *EGFR*-mutated NSCLC and are significantly connected to reduced PFS and OS in patients receiving first-line *EGFR*-TKIs [29]. This negative effect may result from the loss of tumor suppressor function, leading to increased genomic instability and resistance to targeted therapies.

## 5. Conclusions

In conclusion, this study offers valuable insights into the molecular landscape of NSCLC, especially regarding *EGFR* mutations and PD-L1 expression. Although PD-L1 positivity did not show a statistically significant effect on overall survival in this cohort, the link between *EGFR* mutation status and survival outcomes highlights the importance of personalized therapy in NSCLC. The improved survival outcomes associated with TKI therapy further underscore the vital role of targeted treatments in enhancing patient prognosis. Future research with larger sample sizes and longer follow-up periods is needed to better understand the complex relationship between molecular changes, immune response, and treatment results in NSCLC.

## Figures and Tables

**Figure 1 medicina-61-01467-f001:**
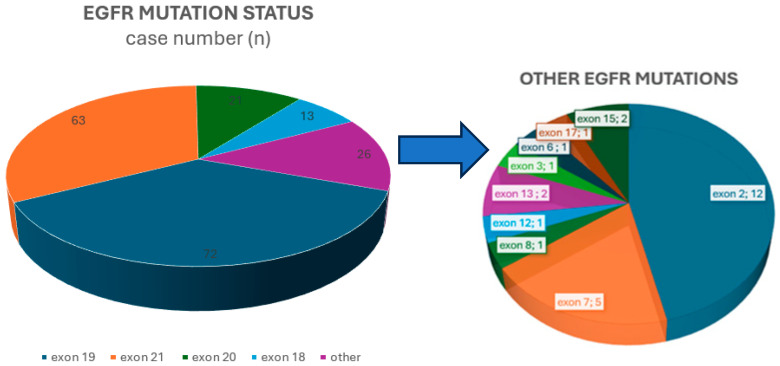
Distribution of *epidermal growth factor receptor* (*EGFR*) mutations.

**Figure 2 medicina-61-01467-f002:**
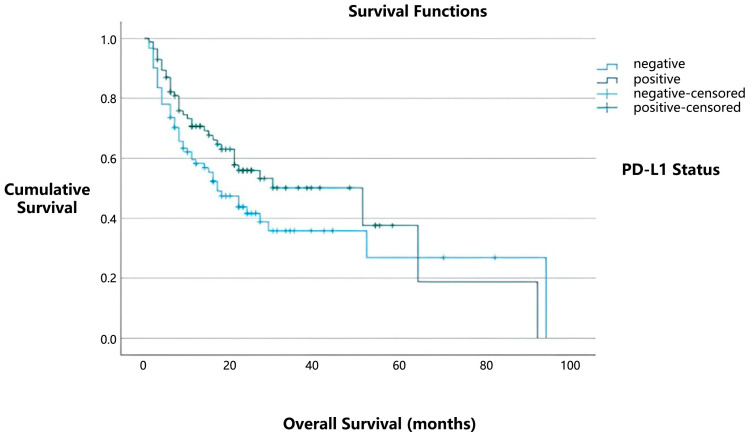
Kaplan–Meier survival curves illustrating the association between PD-L1 expression status and overall survival.

**Figure 3 medicina-61-01467-f003:**
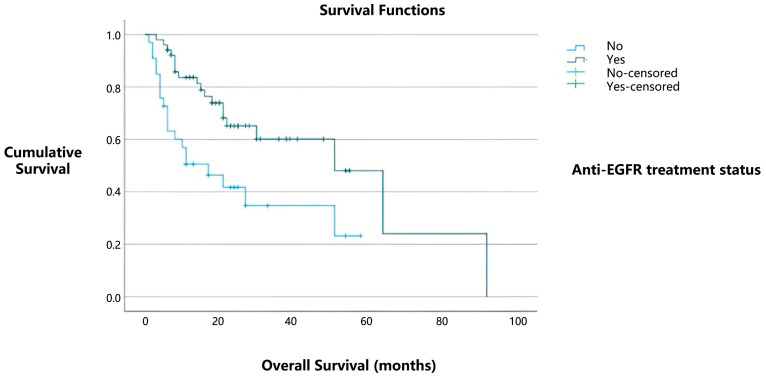
Kaplan–Meier survival curves comparing overall survival in PD-L1-positive patients according to tyrosine kinase inhibitor (TKI) therapy status.

**Figure 4 medicina-61-01467-f004:**
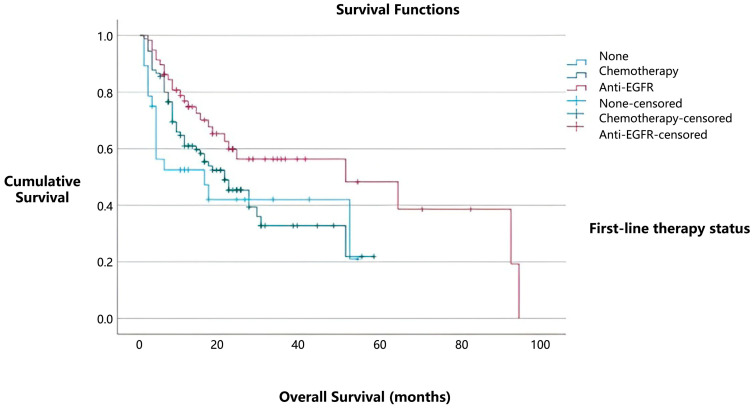
Kaplan–Meier survival curves illustrating the relationship between treatment type and overall survival. Patients who received TKI therapy had the longest median survival, followed by those who underwent chemotherapy, while untreated patients experienced the shortest survival.

**Table 1 medicina-61-01467-t001:** Association between *epidermal growth factor receptor* (*EGFR*) mutation subtypes and some clinicopathological factors.

Features	Exon 19	Exon 20	Exon 21	Exon 18	Exon Other
**Age**					
<65	48.6%	38.1%	38.1%	23.1%	34.7%
≥65	51.4%	61.9%	61.9%	76.9%	65.3%
**Sex**					
Female	63.8%	38%	55.5%	15.3%	7.7%
Male	36.2%	62%	44.5%	84.7%	92.3%
**Histological subtype**					
Adenocarcinoma	97.2%	85.7%	92%	92.3%	65.3%
SCC	2.8%	14.3%	8%	7.7%	34.7%
**Concurrent mutation**					
p53	45.8%	47.6%	46%	30.7%	30.7%
PIK3CA	6.9%	19%	4.7%	7.6%	3.8%
PTEN	2.7%	4.7%	1.5%	0%	0%
KRAS	2.7%	4.7%	1.5%	7.6%	11.5%
Other	12.5%	14.2%	9.5%	23%	26.9%
**Metastatic sites**					
Bone	44.4%	23.8%	38%	30.7%	15.3%
Liver	8.3%	9.5%	11.1%	7.6%	3.8%
Lymph nodes	9.7%	14.2%	17.4%	15.3%	11.5%
Brain	16.6%	4.7%	14.2%	15.3%	7.6%
Other	21%	47.8%	19.3%	31.1%	69.4%
**PD-L1 status**					
Positive	51.3%	42.8%	34.9%	61.5%	61.5%
Negative	48.7%	57.2%	65.1%	38.5%	38.5%
**TIL status**					
Present	87.5%	100%	82.5%	92.3%	84.6%
Absent	12.5%	0%	17.5%	7.7%	15.4%

**Table 2 medicina-61-01467-t002:** Multivariate Cox regression analysis of factors affecting overall survival.

	Multivariate Analysis		
Factor	Hazard Ratio (log Rank)	95% CI	*p* Value
Gender (male/female)	1.047	0.652–1.682	0.848
Age (≥65/<65)	1.183	0.746–1.874	0.45
Performance status (Eastern Cooperative Oncology Group (ECOG)) (≥2/0–1)	1.718	1.062–2.779	0.027 *
Tyrosine kinase inhibitor treatment status (No/Yes)	3.313	2.050–5.356	<0.01 *
Stage (advanced/early)	2.319	1.097–4.903	0.028 *

* Indicates a statistically significant result (*p* < 0.05).

## Data Availability

Data available on request due to restrictions (e.g., privacy, legal or ethical reasons): The data presented in this study are available on request from the corresponding author due to patient confidentiality and ethical restrictions.

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
