# Peer review of "Evaluation of PD-L1 Expression and Anti-EGFR Therapy in EGFR-Mutant Non-Small-Cell Lung Cancer"

_medicina, 2025, doi:10.3390/medicina61081467_

Round 1
Reviewer 1 Report
Comments and Suggestions for Authors
This article does not provide or add any new findings or value toward the understanding of oncogenic gene alterations and their interplay with immune biomarkers in NSCLC, other than confirming what is already known about EGFR gene alterations in patients with metastatic NSCLC and their low response rates for immune checkpoint inhibitors in patients with mutated EGFR and ALK variants than in patients without oncogenic drivers.
Some other areas that need to be clarified:
Major:
- Incorrect conformity to scientific standards. Lack of differentiation between genes and proteins. The genes should either be italicized or underlined, whereas the protein should be written plain text. This article has all genes in capital letters, which are standard convention.
- How was independent assessment of tumor biomarkers performed if there was only 1 pathologist: “A pathologist examined the formalin-fixed paraffin-embedded (FFPE) tissues of patients with non-small cell lung carcinoma and chose the most suitable block for investigation” (section 2.1, line 6)
- Section 2.4: “Following the identification of EGFR mutations, patients demonstrating a favorable clinical and radiological response to chemotherapy were allowed to continue with their existing regimen. Conversely, in patients who did not show a marked response or in whom targeted therapy was deemed more appropriate, chemotherapy was discontinued and EGFR tyrosine kinase inhibitor (TKI) therapy was initiated. “
So basically in this retrospective study, symptomatic patients were initially treated with systemic chemotherapy in absence of availability of oncogenic driver gene alterations?
How were these patients defined?
Based on what parameters (in addition to the detection of oncogenic driver gene alterations) that patients had targeted therapies initiated instead of continuing systemic chemotherapy?
Although the authors did mention the addition of EGFR inhibitors in patients with EGFR exon 19 or 21 gene alterations, it was not clear whether any selection was performed based on the above.
Furthermore, did these patients start without immunotherapy? Then they had immunotherapy added because of PD-L1 or TPS score?
- Section 2.4: “In cases where the T790M mutation was detected, osimertinib was administered as the subsequent line of treatment”. Why was osimertinib not added in patients with exon 19 or 21 gene alterations despite its proven clinical utility? How would this impact overall survival?
- Page 8: “Notably, patients harboring EGFR mutations who were treated with TKIs exhibited the most favorable survival outcomes, with a median overall survival of 51 months (95% CI) (p = 0.033).” How about other co-occurring or secondary mutations, such as MET gene alterations ? Are they only EGFR mutated?
Author Response
POINT BY POINT RESPONSE TO COMMENTS AND SUGGESTIONS OF REVIEWER-1:
General Comment: This article does not provide or add any new findings or value toward the understanding of oncogenic gene alterations and their interplay with immune biomarkers in NSCLC, other than confirming what is already known about EGFR gene alterations in patients with metastatic NSCLC and their low response rates for immune checkpoint inhibitors in patients with mutated EGFR and ALK variants than in patients without oncogenic drivers.
General Response: We recognize the reviewer’s concern that this article confirms known findings regarding EGFR-mutated NSCLC and their low response to immune checkpoint inhibitors compared to patients without oncogenic drivers. While our main goal was not to propose new mechanisms, this study provides real-world evidence from our region, highlighting treatment patterns and survival outcomes within the context of local clinical practice and resource limitations. We have clarified this in the revised discussion section.
Comment 1: Incorrect conformity to scientific standards. Lack of differentiation between genes and proteins. The genes should either be italicized or underlined, whereas the protein should be written plain text. This article has all genes in capital letters, which are standard convention.
Response 1: As the reviewer suggests, the gene names have been corrected to follow standard scientific conventions by italicizing them throughout the manuscript.
Comment 2: How was independent assessment of tumor biomarkers performed if there was only 1 pathologist: “A pathologist examined the formalin-fixed paraffin-embedded (FFPE) tissues of patients with non-small cell lung carcinoma and chose the most suitable block for investigation” (section 2.1, line 6)
Response 2: As the reviewer indicates, we replaced the section that mentions: “A pathologist specialized in thoracic pathology examined hematoxylin and eosin-stained slides of formalin-fixed paraffin-embedded (FFPE) tissue samples from each patient diagnosed with non-small cell lung carcinoma and selected the most representative tumor block for further analysis (section 2.1, paragraph 2, page 4)……Tumor biomarker expression, including EGFR mutations and PD-L1, was independently assessed by two pathologists. Inter-observer variability was evaluated through consensus review, and any disagreements were resolved by joint reexamination to ensure consistent scoring. (section 2.1, paragraph 6, page 5)”
Comment 3: Section 2.4: “Following the identification of EGFR mutations, patients demonstrating a favorable clinical and radiological response to chemotherapy were allowed to continue with their existing regimen. Conversely, in patients who did not show a marked response or in whom targeted therapy was deemed more appropriate, chemotherapy was discontinued and EGFR tyrosine kinase inhibitor (TKI) therapy was initiated. “
- a) So basically in this retrospective study, symptomatic patients were initially treated with systemic chemotherapy in absence of availability of oncogenic driver gene alterations? How were these patients defined? Based on what parameters (in addition to the detection of oncogenic driver gene alterations) that patients had targeted therapies initiated instead of continuing systemic chemotherapy?
- b) Although the authors did mention the addition of EGFR inhibitors in patients with EGFR exon 19 or 21 gene alterations, it was not clear whether any selection was performed based on the above.
- c) Furthermore, did these patients start without immunotherapy? Then they had immunotherapy added because of PD-L1 or TPS score?
Response 3: While the reviewer questioned the parameters for choosing targeted therapies or systemic chemotherapy, we address these questions in the “Systemic treatment approach” subtitle in the Materials and Methods section in detail.
- Initially, systemic chemotherapy was administered to symptomatic patients before the results of next-generation sequencing (NGS) were available for detecting oncogenic driver gene mutations. During this period, platinum-based chemotherapy regimens were the standard first-line treatment. Once NGS results were obtained, patients with EGFR mutations were switched to targeted therapy with tyrosine kinase inhibitors (TKIs). This approach reflects current clinical practice, where delays in molecular diagnostics often result in the initiation of empirical chemotherapy in symptomatic patients. Adjusting therapy based on molecular findings helps optimize treatment effectiveness and patient outcomes. Therefore, the absence of TKI therapy in some EGFR-mutant cases should not be viewed as non-compliance with guidelines but as a reflection of real-world treatment dynamics shaped by diagnostic timelines, clinical urgency, and drug availability.
Patients were classified as symptomatic based on their clinical presentation at diagnosis, which included respiratory symptoms (e.g., cough, dyspnea, hemoptysis), weight loss, fatigue, or radiologically significant tumor burden. Additionally, performance status (ECOG) and clinical urgency were considered.
The decision to switch to targeted therapy instead of continuing systemic chemotherapy was made after EGFR mutations were identified through NGS. In addition to the presence of actionable mutations, treatment options were based on several clinical factors, including the patient's response to initial chemotherapy (assessed through imaging and clinical improvement), chemotherapy tolerability, performance status, comorbidities, and physician judgment. Patients who exhibited poor or no response to chemotherapy, or who experienced significant treatment-related toxicity, were prioritized for switching to EGFR tyrosine kinase inhibitor (TKI) therapy once EGFR mutations were confirmed.
Chemotherapy regimens were selected based on the tumor's histological subtype and individual patient factors, such as age, performance status, and comorbidities. Doublet chemotherapy protocols included combinations like pemetrexed with cisplatin or carboplatin, paclitaxel with cisplatin or carboplatin, gemcitabine with cisplatin or carboplatin, and docetaxel with cisplatin. For some patients, single-agent chemotherapy regimens such as vinorelbine or docetaxel were used. (Material and Methods, Systemic Treatment Approach, Paragraphs 1-4, Page 6-7)
- All patients with confirmed EGFR mutations, including exon 19 deletions and exon 21 L858R mutations, were evaluated for EGFR TKI therapy once molecular results were available. The decision to start or switch to EGFR-targeted therapy was primarily based on the presence of the actionable mutation but also took into account clinical factors such as response to previous chemotherapy, performance status, comorbidities, and tolerance to systemic therapy. Therefore, patient selection was tailored individually, and not all patients immediately transitioned to TKI therapy after EGFR mutation detection. The treatments included erlotinib, gefitinib, afatinib, and osimertinib (the latter in cases with the T790M resistance mutation). (Material and Methods, Systemic Treatment Approach, Paragraph 5, Page 7)
- No patient in our cohort received immunotherapy as a first-line treatment. At the time of initial diagnosis and treatment initiation, immunotherapy was not commonly used in patients with actionable EGFR mutations, as per national guidelines and standard practices. PD-L1 expression (measured by Tumor Proportion Score, TPS) was assessed retrospectively and was included in survival analysis. When used, immunotherapy was typically introduced in later lines of therapy after TKI resistance or in cases without targetable mutations. (Material and Methods, Systemic Treatment Approach, Paragraph 6, Page 7)
Comment 4: Section 2.4: “In cases where the T790M mutation was detected, osimertinib was administered as the subsequent line of treatment”. Why was osimertinib not added in patients with exon 19 or 21 gene alterations despite its proven clinical utility? How would this impact overall survival?
Response 4: Thanks to the reviewer for pointing out that Osimertinib was not added to patients with exon 19 and 21 gene alterations. To answer for it, “In patients who experienced disease progression under TKI therapy and had an estimated life expectancy of more than three months, additional molecular testing was recommended to identify secondary resistance mutations, including the T790M mutation. When the T790M mutation was detected, osimertinib was administered as the next line of treatment. It is important to note that although osimertinib has shown clinical benefit as a first-line agent in patients with EGFR exon 19 deletions or exon 21 L858R mutations, most patients in this study initially received first- or second-generation TKIs (such as erlotinib, gefitinib, or afatinib). This was due to the retrospective nature of the study and the treatment period, during which osimertinib was not routinely approved or reimbursed as a first-line therapy in the national treatment guidelines. As a result, the use of osimertinib was mainly limited to subsequent lines after detecting resistance mutations. These treatment decisions were made based on drug availability, institutional protocols, and the active national regulatory approvals during the study period.”(Material and Methods, Systemic Treatment Approach, Paragraph 7, Page 7-8)
Comment 5: Page 8: “Notably, patients harboring EGFR mutations who were treated with TKIs exhibited the most favorable survival outcomes, with a median overall survival of 51 months (95% CI) (p = 0.033).” How about other co-occurring or secondary mutations, such as MET gene alterations ? Are they only EGFR mutated?
Response 5: As the reviewer indicated the other gene alterations, such as the MET gene alteration, led us to revise the sentence for clarification.
“In our study, the most common concurrent mutation with EGFR was TP53, found in 78 of 176 cases (44.3%). TP53 was followed by PIK3CA in 13 of 176 (7.4%), KRAS in 7 of 176 (4%), and PTEN in 4 of 176 (2.3%) mutations. MET alterations, including amplifications or mutations, were not observed in any cases in our series. The prognostic and predictive significance of TP53 co-mutations in EGFR-mutant NSCLC is well supported by the literature. Among the 78 patients with both EGFR and TP53 mutations, 51 (65.3%) received TKI therapy. The average overall survival was 42 months for patients with TP53 mutations and 49 months for those without, although this difference was not statistically significant (p = 0.924).” (Results, Paragraph 18, Page 15)

Reviewer 2 Report
Comments and Suggestions for Authors
This is a well-organized, comprehensive retrospective analysis of 176 EGFR-mutant NSCLC cases exploring the relationship between PD-L1 expression and anti-EGFR therapy outcomes. The manuscript addresses a timely and clinically relevant topic, particularly important given the growing need to personalize therapy in lung cancer. The combination of molecular profiling (NGS), PD-L1 IHC scoring, and clinical outcome correlation is valuable.
- The language throughout the manuscript requires substantial editing to improve clarity, grammar, and scientific tone.
- The reported association between age and EGFR mutation is incorrect and contradicts the existing literature. EGFR mutations, particularly classical ones, are typically more common in younger patients, not older individuals. The current conclusion is misleading.
- The Introduction fails to reflect current clinical practice, particularly the use of perioperative EGFR-TKI and immunotherapy in early-stage NSCLC. The discussion appears outdated and uninformed about major recent developments in treatment standards.
- The final sentence of the abstract is disconnected from the manuscript findings. It draws a conclusion that is not supported by the presented results and should be revised or removed.
- Variant allele frequency (VAF) was not reported, which is a critical omission. The manuscript should include VAFs for detected mutations to enhance interpretability and reproducibility.
- The discussion should include a more quantitative and critical comparison with Phase III clinical trials in the literature, citing survival data and treatment outcomes from landmark studies.The current discussion summarizes the results but could engage more critically with conflicting literature
- The disease stages of the patients are not clearly defined or reported. Accurate staging is essential for interpreting treatment decisions, survival outcomes, and mutation correlations. The omission of TNM or clinical stage information weakens the clinical relevance and interpretability of the findings.
- The immunotherapy mechanism section in the Introduction is unnecessary and should be removed, as it is not directly relevant to the study's focus and detracts from the main objective.
- PD-L1 expression may vary by tissue site, which is not addressed. It is unclear whether the samples used were obtained via EBUS, surgical resection, or from metastatic sites such as liver, lymph nodes, or brain. This should be clearly stated and discussed.
- The description of osimertinib use in EGFR-mutated cases is inconsistent with current international guidelines. The explanation should clarify that deviations from guideline-based use were due to local reimbursement restrictions in the country, not clinical evidence.
- It is well-known that EGFR mutations are more frequent in females, adenocarcinoma histology, and younger patients. These well-established associations are restated as novel findings. Moreover, there is a contradiction in presenting some of these as consistent while others (e.g., age) contradict prior data.
- It is unclear why certain patients with EGFR mutations did not receive TKI therapy. This omission must be addressed, as it significantly affects the validity of survival analyses.
- The discussion lacks a robust comparison with existing literature. The manuscript should contextualize its findings within the broader field and highlight both consistencies and discrepancies.
- The manuscript does not mention how many patients received immunotherapy or describe their response. Given the relevance of PD-L1 expression, this aspect should be explicitly analyzed and discussed.
- The manuscript states SP263 clone was used, but does not discuss inter-assay variability (SP263 vs 22C3, etc.), which limits external generalizability.
- It would enhance clarity to state explicitly whether scoring was performed by one or multiple pathologists and whether any digital pathology or inter-observer validation was done.
- Co-mutations: TP53 was common (44.3%), but the manuscript does not explore how TP53 co-mutations affected TKI response or survival—this is a known prognostic marker and should be discussed.
- Which EGFR-TKI (1st, 2nd, 3rd gen) was most commonly used? This is critical, especially since osimertinib has superior CNS penetration and overall survival benefit.
- Since high PD-L1 expression might suggest benefit from immunotherapy, more discussion is needed on whether any patients received immune checkpoint inhibitors (even off-label or later lines).
- As a primarily medical study, the surgical aspect is understandably limited. However, since some patients had early-stage disease or lymph node-only metastasis, it would be helpful to note how many (if any) underwent surgery, or if surgery was ever considered or performed before progression.
- Were any tissue samples surgical resections vs biopsies only? This impacts PD-L1 scoring accuracy and the ability to detect co-mutations.
- Please clarify how PS and comorbidities were accounted for to compare OS analysis.
- Ensure all figure legends and tables are self-explanatory (e.g., clarify abbreviations in Table 1).
The language throughout the manuscript requires substantial editing to improve clarity, grammar, and scientific tone.
Author Response
POINT BY POINT RESPONSE TO COMMENTS AND SUGGESTIONS OF REVIEWER-2
First of all, thank you for your review process for this manuscript.
Comment 1: The language throughout the manuscript requires substantial editing to improve clarity, grammar, and scientific tone.
Response 1: The manuscript has been thoroughly revised to enhance clarity, grammar, and scientific tone throughout, as the reviewer recommended.
Comment 2: The reported association between age and EGFR mutation is incorrect and contradicts the existing literature. EGFR mutations, particularly classical ones, are typically more common in younger patients, not older individuals. The current conclusion is misleading.
Response 2: Thank you to the reviewer for their comment. To clarify the conflicting literature data mentioned by the reviewer, we have added two paragraphs to the Discussion section to support our cohort data.
“In our cohort, the average age was 66.23 years, with patients ranging from 40 to 90 years old, and most (60.8%) were aged 65 or older. While previous studies have generally reported a higher prevalence of classical EGFR mutations among younger patients, our findings reveal a statistically significant association between older age and EGFR exon 19 mutations [15]. This difference may result from population-specific factors, including environmental exposures, genetic backgrounds, or biases inherent in retrospective studies. It is also possible that variations in study design, sample size, and regional demographics contribute to these differing results. Therefore, although younger age is often considered characteristic of EGFR-mutant NSCLC, our data emphasize the importance of considering broader patient demographics and suggest that EGFR mutations, especially exon 19 deletions, may also be common among older populations.
Our findings align with those of Tang et al., who reported a higher prevalence of EGFR mutations in patients diagnosed after the age of 50. This supports the idea that older age may significantly influence EGFR mutation status in NSCLC patients. It highlights the need to include a broad age range in future studies and to consider age-related biological and environmental factors that could impact mutation rates. Larger, prospective studies are necessary to better understand the complex relationship between age and EGFR mutation status across different populations [16].”(Discussion, Paragraphs 3-4, Page 16)
Comment 3: The Introduction fails to reflect current clinical practice, particularly the use of perioperative EGFR-TKI and immunotherapy in early-stage NSCLC. The discussion appears outdated and uninformed about major recent developments in treatment standards.
Response 3: Since the reviewer pointed out issues with the Introduction and Discussion sections, we have made the necessary improvements as requested.
“…Phase III clinical trials have shown better outcomes with EGFR-TKIs compared to standard chemotherapy in patients with EGFR-mutant advanced NSCLC [6].
Recent evidence has expanded the use of EGFR-TKIs into the perioperative setting for early-stage NSCLC. The ADAURA trial demonstrated that adjuvant osimertinib significantly improves disease-free survival in patients with resected EGFR-mutant NSCLC, establishing a new standard of care in this field [7]. Furthermore, immunotherapy, especially immune checkpoint inhibitors targeting programmed cell death 1 (PD-1) and its ligand PD-L1, has been integrated into neoadjuvant and adjuvant treatment plans, thereby enhancing long-term outcomes and transforming the management of early-stage NSCLC [8,9]. (Introduction, Paragraphs 2-3, Page 3)
…
Despite the success of PD-1/PD-L1 inhibitors in NSCLC, the interaction between EGFR mutations and PD-L1 expression remains complex and not fully understood. Preclinical studies demonstrate that mutant EGFR increases PD-L1 expression, whereas EGFR-TKI therapy may decrease PD-L1 levels, suggesting a dynamic relationship that affects treatment response [11]. Clinical studies on PD-L1 as a predictive and prognostic biomarker in EGFR-mutant NSCLC treated with EGFR-TKIs have shown mixed results [12], emphasizing the need for further research.”(Introduction, Paragraphs 4, Page 3)
“In our study, none of the patients received immunotherapy as a first-line treatment. Immunotherapy was primarily used in later lines of therapy, often after resistance to EGFR-TKIs developed or when no targetable oncogenic drivers were present. A total of 11 patients received immunotherapy. Although these patients showed a longer median overall survival (30 months) compared to those who did not receive immunotherapy (21 months), the difference was not statistically significant. This may be due to the small sample size and the use of immunotherapy in more advanced disease stages, where tumor burden and performance status could have affected survival outcomes. Additionally, the limited effectiveness of immune checkpoint inhibitors in EGFR-mutant NSCLC, particularly when used after TKI resistance, has been well-documented in previous studies [25]. Despite the trend toward improved survival, our findings suggest that immunotherapy may provide limited benefits in unselected EGFR-mutant populations, particularly when given outside of biomarker-driven treatment strategies. Future prospective studies with larger groups and stratified treatment plans are needed to clarify the role of immunotherapy in this subgroup and to find potential biomarkers of response beyond PD-L1 expression.” (Discussion, Paragraph 11, Page 18)
Comment 4: The final sentence of the abstract is disconnected from the manuscript findings. It draws a conclusion that is not supported by the presented results and should be revised or removed.
Response 4: Thank you very much for your valuable feedback regarding the final sentence in the abstract. We have improved the sentence in the Conclusion section: “EGFR mutation subtypes and PD-L1 expression seem to affect treatment outcomes and survival in NSCLC. The observed links emphasize the potential value of combining molecular and immunological markers to guide therapy choices.”(Abstract, Conclusion, Page 2)
Comment 5: Variant allele frequency (VAF) was not reported, which is a critical omission. The manuscript should include VAFs for detected mutations to enhance interpretability and reproducibility.
Response 5: We would like to kindly point out that in the Materials and Methods section, under “DNA Extraction and EGFR Mutation Assay,” we have specified that in our DNA-NGS analyses performed on FFPE samples, variants with an allele frequency of 5% or higher, along with a reading depth of 500X and a quality score (QUAL) above 200, were considered. (Materials and Methods, DNA extraction and EGFR mutation assay, Paragraph 3, Page 6)
Comment 6: The discussion should include a more quantitative and critical comparison with Phase III clinical trials in the literature, citing survival data and treatment outcomes from landmark studies. The current discussion summarizes the results but could engage more critically with conflicting literature
Response 6: We have significantly revised the Discussion section to include more quantitative comparisons and critical analysis of our findings in relation to major Phase III clinical trials. Specifically, survival outcomes in our cohort were compared with those reported in landmark studies such as FLAURA and ADAURA. For instance, we highlighted that patients receiving osimertinib in our study showed survival outcomes that are consistent with, or better than, those reported in the FLAURA trial (median OS: 38.6 months), indicating the robustness of this approach even in real-world settings. Additionally, we discussed the limited efficacy of immunotherapy in EGFR-mutant NSCLC, aligning with existing literature, and addressed the variability observed in PD-L1 expression as a predictive biomarker. We also included a nuanced discussion of TP53 co-mutations, referencing studies such as Canale et al., and critically acknowledged that, although our findings did not reach statistical significance, the negative prognostic trend is consistent with current evidence. These revisions aim to better contextualize our findings within the broader literature and provide a more comprehensive, balanced interpretation of the data. (Introduction, Paragraph 3, Page 3 and Discussion, Paragraph 12 and last paragraph, Pages 19-20)
Comment 7: The disease stages of the patients are not clearly defined or reported. Accurate staging is essential for interpreting treatment decisions, survival outcomes, and mutation correlations. The omission of TNM or clinical stage information weakens the clinical relevance and interpretability of the findings.
Response 7: Thank you for your insightful comment regarding the patient staging information. To address this, we have now included a clear description of disease stages in the Results section. “Patients were divided into early-stage (Stage I–II) and advanced-stage (Stage III–IV) groups. Out of 176 cases, 20 (11.4%) were classified as early-stage, while 156 (88.6%) were advanced-stage. Lymph node involvement was observed in 22 out of 176 cases (12.5%).” (Results, paragraph 2, Page 9)
Comment 8: The immunotherapy mechanism section in the Introduction is unnecessary and should be removed, as it is not directly relevant to the study's focus and detracts from the main objective.
Response 8: Thank you for your comment. Additionally, another reviewer mentioned this, but instead of opposing it, they suggested supporting this section with a reference. For this reason, I decided to do so rather than remove it. However, the final decision will be up to the editor.
Comment 9: PD-L1 expression may vary by tissue site, which is not addressed. It is unclear whether the samples used were obtained via EBUS, surgical resection, or from metastatic sites such as liver, lymph nodes, or brain. This should be clearly stated and discussed.
Response 9: As the reviewer suggested, we have revised the manuscript to clarify the sampling methods.
“Out of 176 cases, diagnosis was confirmed through surgical resection specimens in 148 patients, while the remaining 28 cases were diagnosed using transbronchial EBUS biopsy or CT-guided transthoracic biopsy. Among the 176 patients in the study, 148 underwent surgical resection (such as wedge resection, lobectomy, or pneumonectomy), primarily for diagnostic purposes or due to early-stage disease. The other 28 patients were diagnosed via EBUS or transthoracic biopsy, typically because of advanced-stage disease or inoperability.
PD-L1 immunohistochemistry and molecular analyses were primarily performed on surgical specimens and primary tumor tissue in most cases, as these samples offer a more complete tissue architecture and higher tumor cellularity. When only biopsy material was available, careful selection of samples with sufficient tumor content was made to ensure the accuracy of both PD-L1 immunohistochemistry and next-generation sequencing (NGS) analyses. Therefore, we do not believe that using biopsy specimens in some patients significantly affected the accuracy of PD-L1 scoring or the detection of genetic mutations. This methodological approach aligns with current international guidelines and is supported by published evidence validating the use of biopsy-based assessments in lung cancer diagnostics.
In patients without resectable disease, PD-L1 expression was evaluated using the most accessible metastatic tissue, such as lymph nodes or lung lesions. Importantly, no PD-L1 tests were performed on liver or brain metastases in this study.” (Materials and Methods, Study Population, Paragraph 3-5, Page 4)
Comment 10: The description of osimertinib use in EGFR-mutated cases is inconsistent with current international guidelines. The explanation should clarify that deviations from guideline-based use were due to local reimbursement restrictions in the country, not clinical evidence.
Response 10: To clarify this issue, we have added detailed information to the “Systemic Treatment Approach” section in the Materials and Methods, Paragraph 7, Page 8. Specifically, we stated that all patients with confirmed EGFR mutations (including exon 19 deletions and exon 21 L858R mutations) were evaluated for EGFR TKI therapy upon availability of molecular results. The decision to initiate or switch to EGFR-targeted therapy was individualized, considering both actionable mutations and clinical parameters such as prior chemotherapy response, performance status, comorbidities, and treatment tolerance.
We also clarified that while osimertinib has demonstrated benefits as a first-line agent, most patients in our retrospective cohort from 2020 to 2023 initially received first- or second-generation TKIs because osimertinib was not routinely approved or reimbursed as a first-line therapy under national guidelines during this period. As a result, osimertinib was mainly used later in treatment, after resistance mutations were identified, particularly T790M. These decisions were shaped by drug availability, institutional protocols, and national regulatory approvals in effect at the time.
We believe this addition addresses concerns about deviations from international guidelines and provides context for the treatment patterns observed in our study.
Comment 11: It is well-known that EGFR mutations are more frequent in females, adenocarcinoma histology, and younger patients. These well-established associations are restated as novel findings. Moreover, there is a contradiction in presenting some of these as consistent while others (e.g., age) contradict prior data.
Response 11: We appreciate your comment regarding the inclusion of well-established associations such as the higher frequency of EGFR mutations in females, younger patients, and adenocarcinoma histology. Our goal was not to present these associations as new findings but to report and confirm the distribution patterns observed in our own cohort. These demographic and histological characteristics were included in the Results section (Paragraph 1, Page 9) for completeness and transparency because they provide context for interpreting mutation frequencies and treatment outcomes within our study population. Furthermore, in the Discussion section (Paragraph 3, Page 16), we aimed to critically compare our findings with those in the existing literature. In doing so, we acknowledged consistencies—such as the predominance of EGFR mutations in adenocarcinoma—and also highlighted areas of divergence, such as the association of exon 19 mutations with older age in our cohort. This was discussed in consideration of possible regional, genetic, and methodological differences. Our objective was to contribute to the understanding of EGFR mutation patterns in diverse populations rather than to restate known facts without context. We have revised the relevant sections to ensure these points are clearly framed as confirmatory and comparative, rather than as new discoveries.
Comment 12: It is unclear why certain patients with EGFR mutations did not receive TKI therapy. This omission must be addressed, as it significantly affects the validity of survival analyses.
Response 12: We thank the reviewer for their comment regarding the administration of TKI therapy in EGFR-mutant patients. In response, we have expanded the Systemic Treatment Approach section of the Materials and Methods (Page 8, last paragraph) to provide a more detailed explanation of our treatment strategy.
Specifically, the decision to initiate TKI therapy was based on the molecular testing results. However, because of the retrospective nature of the study and the time needed to obtain next-generation sequencing (NGS) results, systemic chemotherapy was often started in symptomatic patients before the molecular data were available. This reflects common clinical practice in many centers where delays in molecular diagnostics lead to empirical chemotherapy to manage tumor-related symptoms and prevent rapid clinical decline.
Once EGFR mutations were confirmed, treatment decisions were customized based on clinical factors such as performance status, response to initial chemotherapy, tolerance, presence of comorbidities, and patient preference. Therefore, not all patients immediately transitioned to EGFR-targeted therapy, especially when they responded well to chemotherapy or had limited expected benefit from switching treatments. Additionally, during the study period (2020–2023), the use of osimertinib as a first-line treatment was limited due to national reimbursement restrictions, which further impacted treatment options.
Importantly, these treatment paths mirrored real-world clinical practice rather than diverging from evidence-based guidelines. Therefore, the absence of TKI therapy in some EGFR-mutant cases should not be considered a flaw in the methodology, but instead as an example of real-world variation in diagnostic and treatment timelines.
Comment 13: The discussion lacks a robust comparison with existing literature. The manuscript should contextualize its findings within the broader field and highlight both consistencies and discrepancies.
Response 13: In response to your suggestion, we have expanded the discussion section to offer a more comprehensive comparison of our findings with current and important literature.
Comment 14: The manuscript does not mention how many patients received immunotherapy or describe their response. Given the relevance of PD-L1 expression, this aspect should be explicitly analyzed and discussed.
Response 14: We have revised the Results section to explicitly state the number of patients who received immunotherapy and provided further clarification regarding the timing and context of its use.
“None of the patients in our cohort received immunotherapy as first-line treatment. When it was used, immunotherapy was typically administered later in treatment, either after developing resistance to tyrosine kinase inhibitors (TKIs) or in cases without targetable mutations. A total of 11 patients in our study received immunotherapy. The overall survival of these patients was analyzed statistically. Although the median overall survival was longer for patients who received immunotherapy (30 months) compared to those who did not (21 months), this difference was not statistically significant (p = 0.833).”(Results, Paragraph 12, Page 13)
Comment 15: The manuscript states SP263 clone was used, but does not discuss inter-assay variability (SP263 vs 22C3, etc.), which limits external generalizability.
Response 15: Thank you very much for your valuable comment regarding the use of the SP263 clone for PD-L1 immunohistochemistry and the inter-assay variability between different antibody clones. We conducted the study with the SP263 clone, and the purpose of the study was not to compare PD-L1 clones. However, in the discussion section, we discussed the literature data for both SP263 and other PD-L1 clones.
“PD-L1 expression is a crucial biomarker in NSCLC, guiding the use of immune checkpoint inhibitors, especially anti-PD-1/PD-L1 therapies [23]. In this study, PD-L1 expression was evaluated using the SP263 clone, a widely employed marker in clinical practice. However, it is essential to acknowledge that variability exists among different PD-L1 immunohistochemistry assays, including SP263, 22C3, 28-8, and SP142 clones. Several studies have shown that, although SP263 and 22C3 clones typically demonstrate good agreement, differences in staining intensity, scoring thresholds, and tumor proportion scoring can impact comparability and clinical interpretation across assays [24]. This variability may limit the direct external applicability of our findings to settings that use alternative PD-L1 assays. Therefore, caution should be exercised when applying PD-L1 expression results obtained with SP263 to broader populations or to studies using different clones. Future research should focus on standardizing PD-L1 testing methods and incorporating cross-validation of assays to enhance comparability and strengthen biomarker-driven therapeutic decision-making.” (Discussion, Paragraph 8, Page 17)
Comment 16: It would enhance clarity to state explicitly whether scoring was performed by one or multiple pathologists and whether any digital pathology or inter-observer validation was done.
Response 16: As the reviewer pointed out, we clarified whether scoring was performed by one or multiple pathologists and whether any digital pathology or inter-observer validation was done. (Materials and Methods, PD-L1 expression analysis, whole section, Page 6)
Comment 17: Co-mutations: TP53 was common (44.3%), but the manuscript does not explore how TP53 co-mutations affected TKI response or survival—this is a known prognostic marker and should be discussed.
Response 17: As the reviewer pointed out, it has been discussed in the Results section, last paragraph(Page 15).
“In our study, the most common concurrent mutation with EGFR was TP53, found in 78 of 176 cases (44.3%). TP53 was followed by PIK3CA in 13 of 176 (7.4%), KRAS in 7 of 176 (4%), and PTEN in 4 of 176 (2.3%) mutations. The prognostic and predictive significance of TP53 co-mutations in EGFR-mutant NSCLC is well supported by the literature. Among the 78 patients with both EGFR and TP53 mutations, 51 (65.3%) received TKI therapy. The average overall survival was 42 months for patients with TP53 mutations and 49 months for those without, although this difference was not statistically significant (p = 0.924).”
Comment 18: Which EGFR-TKI (1st, 2nd, 3rd gen) was most commonly used? This is critical, especially since osimertinib has superior CNS penetration and overall survival benefit.
Response 18: We have added detailed information regarding the types of EGFR-TKIs used in our cohort to the Results section, Paragraph 13, Page 13.
“In our study, 58 / 176 (32.9%) patients received tyrosine kinase inhibitor (TKI) therapy as first-line treatment, with a median overall survival of 51 months. Among these 58 patients, 44 received first-generation TKIs (erlotinib or gefitinib), 11 received second-generation TKIs (afatinib), and only three patients received third-generation TKI (osimertinib) as initial treatment. However, additional patients were later switched to osimertinib after the detection of acquired T790M resistance mutations. In total, 19 patients ultimately received osimertinib during their treatment course. Statistical analysis showed that patients who received osimertinib had significantly longer overall survival compared to those who did not. The mean overall survival was 92 months in the osimertinib group versus 21 months in the non-osimertinib group (p = 0.002).”
Comment 19: Since high PD-L1 expression might suggest benefit from immunotherapy, more discussion is needed on whether any patients received immune checkpoint inhibitors (even off-label or later lines).
Response 19: We have provided additional details regarding the use of immunotherapy in our cohort and included this information in the Results section, Paragraph 12, Page 13.
“None of the patients in our cohort received immunotherapy as first-line treatment. When it was used, immunotherapy was typically administered later in treatment, either after developing resistance to tyrosine kinase inhibitors (TKIs) or in cases without targetable mutations. A total of 11 patients in our study received immunotherapy. The overall survival of these patients was analyzed statistically. Although the median overall survival was longer for patients who received immunotherapy (30 months) compared to those who did not (21 months), this difference was not statistically significant (p = 0.833).”
Comment 20: As a primarily medical study, the surgical aspect is understandably limited. However, since some patients had early-stage disease or lymph node-only metastasis, it would be helpful to note how many (if any) underwent surgery, or if surgery was ever considered or performed before progression.
Response 20: As the reviewer suggested, we have clarified the extent of surgical intervention in our cohort.
“Out of 176 cases, diagnosis was confirmed through surgical resection specimens in 148 patients, while the remaining 28 cases were diagnosed using transbronchial EBUS biopsy or CT-guided transthoracic biopsy. Among the 176 patients in the study, 148 underwent surgical resection (such as wedge resection, lobectomy, or pneumonectomy), primarily for diagnostic purposes or due to early-stage disease. The other 28 patients were diagnosed via EBUS or transthoracic biopsy, typically because of advanced-stage disease or inoperability.” (Material and Methods, Study Population, Paragraph 3, Page 4)
Comment 21: Were any tissue samples surgical resections vs biopsies only? This impacts PD-L1 scoring accuracy and the ability to detect co-mutations.
Response 21: We address this comment in part in Response 20. Additionally, we have mentioned it in the Materials and Methods, Study Population, Paragraphs 4-5, Page 4.
“PD-L1 immunohistochemistry and molecular analyses were primarily performed on surgical specimens and primary tumor tissue in most cases, as these samples offer a more complete tissue architecture and higher tumor cellularity. When only biopsy material was available, careful selection of samples with sufficient tumor content was made to ensure the accuracy of both PD-L1 immunohistochemistry and next-generation sequencing (NGS) analyses. Therefore, we do not believe that using biopsy specimens in some patients significantly affected the accuracy of PD-L1 scoring or the detection of genetic mutations. This methodological approach aligns with current international guidelines and is supported by published evidence validating the use of biopsy-based assessments in lung cancer diagnostics.
In patients without resectable disease, PD-L1 expression was evaluated using the most accessible metastatic tissue, such as lymph nodes or lung lesions. Importantly, no PD-L1 tests were performed on liver or brain metastases in this study.”
Comment 22: Please clarify how PS and comorbidities were accounted for to compare OS analysis.
Response 22: We clarified this comment in the Results section, paragraphs 16 and 17, and summarized it in Table 2, Pages 14-15.
“In our study, the Eastern Cooperative Oncology Group (ECOG) performance status at diagnosis was recorded for all patients and included in the overall survival (OS) analysis. ECOG status was not assessed in isolation; instead, it was used as a covariate in a multivariate Cox proportional hazards regression model, along with age, sex, disease stage, and TKI treatment status. This approach enabled us to assess the independent prognostic impact of ECOG status on survival, adjusting for relevant clinical factors.
In the multivariate Cox regression analysis, performance status (ECOG ≥2), lack of TKI therapy, and advanced stage at diagnosis were identified as independent predictors of poorer overall survival. Specifically, patients with an ECOG performance status of≥2 had a 1.72-fold increased risk of death compared to those with an ECOG performance status of 0–1 (HR: 1.718, 95% CI: 1.062–2.779, p = 0.027). The absence of TKI treatment was associated with a significantly higher risk of mortality (HR: 3.313, 95% CI: 2.050–5.356, p < 0.01). Additionally, patients with advanced-stage disease faced a 2.3-fold increased risk of death compared to those with early-stage disease (HR: 2.319, 95% CI: 1.097–4.903, p=0.028). Gender and age were not significantly associated with survival in the multivariate model. These findings are summarized in Table 2.”
Comment 23: Ensure all figure legends and tables are self-explanatory (e.g., clarify abbreviations in Table 1).
Response 23: As the reviewer requested, we ensured that the figure legends and tables are self-explanatory. We also expanded the abbreviations in the figure legends.

Reviewer 3 Report
Comments and Suggestions for Authors
This single-center, retrospective study interrogates 176 Turkish non-small cell lung cancer cases diagnosed between 2020 and 2024 to clarify how EGFR‐mutation subtypes interact with PD-L1 expression and systemic therapy. EGFR alterations were dominated by exon 19 deletions (40.9 %) and exon 21 L858R substitutions (35.8 %). PD-L1 positivity (TPS ≥ 1 %) was seen in 48.3 % of tumours, yet high expression (TPS ≥ 50 %) in only 23.5 %; exon 21 mutations correlated with PD-L1 negativity, whereas exon 19 lesions were linked to older age, female sex and bone metastasis. Survival analyses showed median overall survival of 51 months for patients receiving tyrosine-kinase inhibitors versus 21 months with chemotherapy and 16 months with no systemic treatment; PD-L1-positive patients benefitted most from TKIs. The authors conclude that combined molecular and immune profiling refines prognostication and supports preferential use of EGFR-TKIs in EGFR-mutant NSCLC irrespective of PD-L1 status.
Comments:
1. The third-paragraph statements on immune-checkpoint regulation in EGFR-mutant NSCLC “Immune checkpoints are regulated by … EGFR-mutant NSCLC patients” needs one or two contemporary references.
2. Please cite a guideline or justify using <1 % / ≥1 % for PD-L1 as the dichotomous cut-off.
3. How many patients received each doublet or single-agent regimen? Provide an n (%) for every group so the reader can follow the survival analysis.
4. Please add smoking history, TNM-stage, lymph-node involvement, and whether cases were de-novo vs. relapsed.
5. This reviewer thinks the first mention of Table 1 should be at the start of the Results paragraph that discusses age/sex/histology.
6. All Figures are low quality.
7. List how many patients harbored co-mutations of EGFR, TP53, PIK3CA, KRAS, PTEN, etc., and clarify whether co-mutations were included in statistical models.
8. Methods state all 176 cases were EGFR-mutant, yet the Abstract/Results say “EGFR mutations were present in 47.7 %”. Either correct the percentage to 100 % or clarify that 176 is the whole NSCLC cohort and only 84 were mutant.
9. Counts for PD-L1-positive patients sum to 76 (37 + 9 + 22 + 8) but 85 positives are reported. Please account for the missing nine “other exon” cases.
10. Age, stage, performance status, and treatment line should be entered into a Cox model; univariate Kaplan-Meier curves are insufficient to support causal statements.
11. Please add raw n with every % (e.g., “adenocarcinoma 156/176 (88.6 %)”) to let readers track arithmetic.
12. Specify software version, two-tailed p-value threshold, and whether proportional-hazards assumptions were tested.
13. Please correct typos: “propotion” → “proportion”, “tumour propotion score” → “tumor-proportion score”.
Author Response
POINT BY POINT RESPONSE TO COMMENTS AND SUGGESTIONS OF REVIEWER-3
First of all, thank you for your review process for this manuscript.
Comment 1: The third-paragraph statements on immune-checkpoint regulation in EGFR-mutant NSCLC “Immune checkpoints are regulated by … EGFR-mutant NSCLC patients” needs one or two contemporary references
Response 1: Thank you for bringing this matter to our attention. We agree with your opinion, so we have cited the following recent and relevant publication to support the sentence:
"Immune checkpoint regulation involves co-stimulatory and co-inhibitory molecules that modulate tumor-infiltrating lymphocytes (TILs)." The reference has been added to the manuscript as Reference 10:
Franzese O. Tumor Microenvironment Drives the Cross Talk Between Co-Stimulatory and Inhibitory Molecules in Tumor-Infiltrating Lymphocytes: Implications for Optimizing Immunotherapy Outcomes. Int J Mol Sci. 2024;25(23):12848. https://doi.org/10.3390/ijms252312848.
Comment 2: Please cite a guideline or justify using <1 % / ≥1 % for PD-L1 as the dichotomous cut-off.
Response 2: As the reviewer suggested, we have clarified in the Materials and Methods section under “PD-L1 Expression Analysis” that the use of the <1% / ≥1% cutoff for PD-L1 expression classification is based on established international clinical guidelines. Specifically, we referenced the IASLC Atlas of PD-L1 Testing in Lung Cancer, which recognizes TPS ≥1% as a clinically meaningful threshold for PD-L1 positivity in non-small cell lung cancer. The relevant sentences and citation have been added to the manuscript as Reference 13.
Comment 3: How many patients received each doublet or single-agent regimen? Provide an n (%) for every group so the reader can follow the survival analysis.
Response 3: Thanks for the comment. Therefore, we have added detailed information regarding the chemotherapy regimens in the Results section. Chemotherapy was administered to 90 cases (51.1% of the total), who demonstrated a median overall survival of 21 months. Among these patients receiving first-line chemotherapy, 4 (4.4%) were treated with single-agent chemotherapy, 61 (67.8%) received doublet chemotherapy, and the remaining 25 (27.8%) received combination chemotherapy consisting of three or more agents. We believe this addition provides clearer insight into the distribution of chemotherapy regimens and facilitates a better understanding of the survival analysis. (Results, Paragraph 14, Page 14)
Comment 4: Please add smoking history, TNM-stage, lymph-node involvement, and whether cases were de-novo vs. relapsed.
Response 4: As the reviewer requests, we have added the following information to the Results section: Patients were divided into early-stage (Stage I–II) and advanced-stage (Stage III–IV) groups. Out of 176 cases, 20 (11.4%) were classified as early-stage, while 156 (88.6%) were advanced-stage. Lymph node involvement was observed in 22 out of 176 cases (12.5%). All cases were de novo. However, due to incomplete clinical data, smoking history was unavailable for some patients, so a definitive percentage regarding smoking status could not be reported. (Results, Paragraph 2, Page 9)
Comment 5: This reviewer thinks the first mention of Table 1 should be at the start of the Results paragraph that discusses age/sex/histology.
Response 5: Thank you to the reviewer for their comment. Therefore, we placed Table 1 not at the beginning of the results paragraph that discusses age, sex, and histology. While Table 1 pertains to EGFR gene mutation subtypes and some clinicopathological parameters, we have positioned it in the most appropriate location, which is under the paragraph that mentions these parameters.
Comment 6: All Figures are low quality.
Response 6: We have reviewed all figures and increased their resolution to ensure better quality and clarity in the revised version, as the reviewer recommended.
Comment 7: List how many patients harbored co-mutations of EGFR, TP53, PIK3CA, KRAS, PTEN, etc., and clarify whether co-mutations were included in statistical models.
Response 7: In our cohort, although all cases had EGFR mutations, co-occurring mutations were also present. As previously noted in the Results section, the most common concurrent mutation was TP53, found in 44.3% of cases, followed by PIK3CA (7.4%), KRAS (4%), and PTEN (2.3%) mutations.
Co-mutations were evaluated, and their frequencies are reported; however, due to the limited number of cases with specific co-mutations, they were not included as separate variables in the statistical models to prevent overfitting. Instead, the analysis concentrated on the primary mutations of interest. Additionally, another reviewer's recommendation addressed this issue, leading us to evaluate TP53 (the most frequent co-mutation and its association with poor prognosis), and we conducted survival analysis. (Results, last paragraph, page 15)
Comment 8: Methods state all 176 cases were EGFR-mutant, yet the Abstract/Results say “EGFR mutations were present in 47.7 %”. Either correct the percentage to 100 % or clarify that 176 is the whole NSCLC cohort and only 84 were mutant.
Response 8: We appreciate the reviewer’s observation. The statement in the Abstract has been corrected to avoid the misleading implication that only 47.7% of the cases were EGFR-mutant. As clarified in the Methods section, all 176 cases included in this study harbored EGFR mutations. The revised sentence in the Abstract now accurately reflects the distribution of EGFR mutation subtypes: “Within the EGFR mutation spectrum, exon 19 deletions were the most common subtype, accounting for 40.9% of cases, followed by the point mutation in exon 21, which occurred in 35.8% of cases. Less frequent alterations, making up 23.3% of all detected mutations, included mutations in exon 18, insertions, and point mutations such as S768I and T790M in exon 20, as well as changes in exon 2, exon 7, and other less frequently affected regions.” (Results, Paragraph 3, Page 9)
Comment 9: Counts for PD-L1-positive patients sum to 76 (37 + 9 + 22 + 8) but 85 positives are reported. Please account for the missing nine “other exon” cases.
Response 9: As noted, the initial counts for PD-L1-positive patients totaled 76; however, the remaining 9 cases with PD-L1 positivity were classified under the “other exons” group. This has now been clarified in the revised Results section. Specifically, the distribution of these nine cases was as follows: “The remaining 9 cases fell into the 'other exons' group: 1 involved exon 17, 2 involved exon 7, 1 involved exon 6, 1 involved exon 8, 2 involved exon 15, and 2 involved exon 2.”(Results, Paragraph 9, Page 11)
Comment 10: Age, stage, performance status, and treatment line should be entered into a Cox model; univariate Kaplan-Meier curves are insufficient to support causal statements.
Response 10: Thank you for your important comment. As the reviewer suggests, a multivariate Cox proportional hazards regression analysis was performed to evaluate the independent prognostic value of variables including age, gender, stage, ECOG performance status, and TKI treatment status. The results showed that poor performance status (ECOG ≥2), advanced stage, and lack of TKI treatment were significantly linked to worse overall survival. These findings are summarized in Table 2.(Results, Paragraph 17, Page 15)
Comment 11: Please add raw n with every % (e.g., “adenocarcinoma 156/176 (88.6 %)”) to let readers track arithmetic
Response 11: As the reviewer suggests, we have corrected each percent to help readers follow the calculations.
Comment 12: Specify software version, two-tailed p-value threshold, and whether proportional-hazards assumptions were tested.
Response 12: As the reviewer requests, we have added these details to the Materials and Methods section: “All data were entered into the SPSS database (IBM SPSS Statistics, version 27.0), and a chi-square test was performed to compare EGFR mutation status, the presence of concomitant mutations, and age data obtained through NGS…..For survival analysis, univariate analysis was performed using the Kaplan-Meier method, and statistical significance was assessed with the log-rank test. Independent prognostic factors for overall survival were evaluated through multivariate Cox proportional hazards regression analysis. The hazard ratio (HR) was used to measure the effect of variables on overall survival. For all variables, 95% confidence intervals (CI) were calculated and reported when applicable. A p-value of less than 0.05 was considered statistically significant.”(Material and Methods, Statistical Analysis, whole section, Page 8)
Comment 13: Please correct typos: “propotion” → “proportion”, “tumour propotion score” → “tumor-proportion score”.
Response 13: As the reviewer requested, we have fixed the typos that were indicated.

Round 2
Reviewer 1 Report
Comments and Suggestions for Authors
The revision is considered appropriate for publication.
Author Response
We would like to thank Reviewer 1 for their valuable time, constructive comments, and insightful suggestions, which have significantly contributed to improving our manuscript.
Reviewer 2 Report
Comments and Suggestions for Authors
the authors has addressed my comments
Author Response
We would like to thank Reviewer 2 for their valuable time, constructive comments, and insightful suggestions, which have significantly contributed to improving our manuscript.
Reviewer 3 Report
Comments and Suggestions for Authors
The authors have done an excellent job revising the manuscript; all substantive concerns have been satisfactorily addressed and the work is now much clearer and more robust. Only one small numerical error remains: the high PD-L1 expression rate is calculated incorrectly—22 / 85 corresponds to 25.9 %, not 23.5 %. Once this figure is corrected wherever it appears, I consider the manuscript suitable for publication.
Author Response
Comments 1: [The authors have done an excellent job revising the manuscript; all substantive concerns have been satisfactorily addressed and the work is now much clearer and more robust. Only one small numerical error remains: the high PD-L1 expression rate is calculated incorrectly—22 / 85 corresponds to 25.9 %, not 23.5 %. Once this figure is corrected wherever it appears, I consider the manuscript suitable for publication.]
Response 1: We would like to thank Reviewer 3 for their valuable time, constructive comments, and insightful suggestions, which have significantly contributed to improving our manuscript. The remaining numerical error that had remained has been changed from 23.5% to 25.9%. (Results Section, Page 11, Line 5)